# Variability in radiocesium activity concentration in growing hardwood shoots in Fukushima, Japan

**Hiroki Itô** [1]*, **Satoru Miura** [2], **Masabumi Komatsu** [2], **Tsutomu Kanasashi** [3], **Junko Nagakura** [2], **Keizo Hirai** [2]

1 Hokkaido Research Center, Forestry and Forest Products Research Institute, Sapporo, Japan, 2 Forestry and Forest Products Research Institute, Tsukuba, Japan, 3 Institute of Environmental Radioactivity, Fukushima University, Fukushima, Japan

* abies.firma@gmail.com

**Data Availability Statement:** All relevant data are within the paper and its Supporting information files.

## Abstract

The radiocesium contamination caused by the Fukushima Daiichi Nuclear Power Plant accident has made it difficult to use coppice woods as bed logs for mushroom cultivation. Evaluating the variability in the radiocesium activity concentration of logs is necessary in order to predict how many coppice woodlands are available for producing mushroom bed logs. To clarify the variability in radiocesium activity concentrations and to estimate the sample size required to estimate these concentrations with sufficient accuracy, we modeled the log-transformed radiocesium activity concentrations in growing shoots of hardwoods. We designed two models: (1) a model with mean concentrations that varied among stands with a standard deviation that was the same among stands, and (2) a model with varying means and standard deviations. We fit the data pertaining to only *Quercus serrata* to both models and calculated the widely applicable information criterion values. Consequently, we adopted the simpler model (1). Applying the selected model to data for all species, we examined the relationship between the number of measurement individuals and the predictive distribution of the expected concentration. Based on previous recommendations and measurement costs, we proposed that five individuals would be appropriate for estimating radiocesium activity concentration in a stand.

## Introduction

In Japan, coppice woodlands, consisting mainly of oaks, have long been managed for obtaining wood for fuel, charcoal, bed logs for mushroom cultivation, and for other purposes. Woodlands in Japan are a major component of what is referred to locally as "satoyama" landscape, which are areas that comprised of woodlands, grasslands, paddy fields, ponds, and related land use types [1, 2]. These areas, which are characterized by periodic human disturbance, have relatively high levels of biodiversity [3]. However, although managed coppice woodlands can maintain biodiversity [4], large areas of them have been abandoned due to socio-economic changes [5].

**Funding:** This research was supported by the research program on development of innovative technology grants from the Project of the Bio-oriented Technology Research Advancement Institution (BRAIN) (Grant Number: JPJ007097) and grants from the Forestry and Forest Products Research Institute (201901, 202203).

**Competing interests:** The authors have declared that no competing interests exist.

More than any other areas in Japan, the traditional "satoyama" landscape is most prevalent in the southern Tohoku and northern Kanto regions of Japan. In Fukushima Prefecture, which is located in southern Tohoku, large area of coppice woodlands have been managed to produce the bed logs that are used for mushroom cultivation [6, 7]. For this purpose, oak *Quercus serrata* and other hardwood tree species have been cultivated in the coppice woodland. The production of these bed logs previously benefited forestry activities and the regional economy of the area, as well as supporting the biodiversity of the region.

However, the Fukushima Daiichi Nuclear Power Plant accident, which occurred in March 2011, polluted the area with radioactive materials. Most of the fallout consisted of the radiocesium isotopes, $^{134}$Cs and $^{137}$Cs. Though, as a result of its longer 30-year half-life, only $^{137}$Cs remains in the environment. This contamination has had an enormous impact on the communities and industries in the region, including the production of bed logs for mushroom cultivation in the oak coppices.

In order to meet the required safety standards limits for food, 100 Bq kg$^{-1}$, bed logs used for mushroom cultivation must have a radioactivity concentration lower than 50 Bq kg$^{-1}$ [8]. This is because the maximum transfer factor from the bed log to cultivated mushrooms has been estimated as being approximately two [7]. Resuming bed log production is considered to be necessary for the forestry and regional economy and would also contribute towards maintaining the biodiversity of the region.

Recently, studies have been conducted to facilitate the resumption of coppice management. For example, Kanasashi et al. [9] established a strong and negative correlation between the exchangeable potassium present in soil and the radiocesium ($^{137}$Cs) activity concentrations in the growing shoots of *Q. serrata*. The observation of a constant relationship between radiocesium activity concentration of growing shoots (new branches) and that in woody stem means that the concentration in growing shoots are well suited for use as a proxy for the concentration in woody stems [10, 11]. Based on this assumption, coppices situated on the potassium-rich soils may be well suited for the production of bed logs with low radiocesium contamination even in areas that were experienced radioactive fallout.

The application of potassium fertilizer to reduce the absorption of radiocesium is a widely used measure in agricultural fields [12]. In woodlands, which are considerably larger than agricultural lands, searching for coppices with low radiocesium activity concentrations could also be a promising measure for sourcing bed logs with radiocesium activity concentrations that are within permissible limits. Indeed, for mushroom cultivation on sawdust media, the relationship between mushroom fruiting body and medium radiocesium activity concentrations have been clarified [13, 14]. To accurately predict the proportion of available logs, estimating not only the mean, but also the variances in the radiocesium activity concentrations present in the coppice stems is crucial. It has been shown that a definitive range exists in the radiocesium activity concentrations of growing shoots within and among stands of coppices and plant species [9]. However, this variance is seldom known a priori [15, 16]. Moreover, if the variance in the radiocesium activity concentrations varies among stands, then it will be difficult to accurately predict the concentrations from small sample sizes. It is therefore necessary to clarify the distribution of the variance in the radiocesium activity concentration within and among coppice stands. In addition, the variance would be important to determine the required sample size to predict the concentration.

Two questions arise regarding the measurement of radiocesium activity concentration in growing shoots: (1) Can the variances of the measurement values of the radiocesium activity concentration in growing shoots be regarded as constant or variable among different stands? (2) What number of individual samples are required to measure the stand-level distribution of radiocesium activity concentration? In this study, we evaluated the stand-level variation in

radiocesium activity concentrations in the growing shoots of coppice species, and predicted the concentration distribution within a stand. The findings will enable the estimation of the mean radiocesium activity concentration in growing shoots within a specified area, thereby ensuring that the concentration distribution is under a predetermined threshold.

## Materials and methods

### Data collection

We established 40 study stands in Miyakoji, Tamura City, Fukushima Prefecture, Japan, and sampled growing shoots of hardwood species in each stand between December 2016 and March 2017. The radiocesium activity concentration is known to be stable during the dormant stage (November to April) [17]. In this study, we assumed that the stands comprised areas of several tens of square meters in size. Permission was obtained from the Fukushima Chuo Forestry Co-operative to collect samples from the study area. The *Q. serrata* data used in this analysis were essentially the same as those in Kanasashi et al. [9], except that some measurements were updated. The data file is provided in the supporting information (S1 Table). In addition, other hardwood species that were present in the stands with *Q. serrata*, as well as regenerated coppice sprouts and new plantation, were also included in the analysis. A total of 453 individuals of hardwood tree species were collected. Of these, 418 individuals of five species had radiocesium activity concentrations that could be detected by the method described below were included in the analysis (i.e., 35 individuals (7.7%) were excluded) (Table 1).

The collected samples were processed as described in a previous study [9], and the radioactivity concentrations were measured using a NaI(Tl) scintillation detector (2480 WIZARD[2]; PerkinElmer Japan Co., Ltd., Yokohama, Japan) or high-purity germanium detectors (GEM40P4-76 and GEM-FX7025P4-ST; ORTEC, Oak Ridge, TN, USA). All of the detectors were calibrated using the radioactivity standard gamma volume sources, MX035 and MX033U8PP (Japan Radioisotope Association, Tokyo, Japan). For the NaI(Tl) scintillation detector measurements, samples were placed into 20-mL vials and measured from 2,000 to 172,800 s, after which the gamma-ray spectra were analyzed using the Code Fukushima software package [18]. The detection limit is not constant and depends on the radioactivity concentration and measurement time (S1 Table). The 662 keV peaks of Cs-137 were considered to be a distinct peak when the full width at half maximum (FWHM) was greater than 30 keV and the relative counting error was less than 33.3%; peaks meeting these criteria were used for the analysis. For the germanium semiconductor detector, measurements were performed from 1,800 to 86,400 s, and data with relative counting errors of 33.3% or less were used for analysis. The activity concentrations were decay-corrected to the values on December 1, 2016.

### Statistical modeling

We assumed that the natural logarithm (hereafter written as logarithm) of the radiocesium activity concentration $Y_i$ (Bq kg$^{-1}$) in the growing shoots followed a normal distribution [15,

**Table 1. Number of individuals of each species that were sampled.**

| Species | No. |
| --- | --- |
| *Quercus serrata* Murray | 184 |
| *Cerasus jamasakura* (Siebold ex Koidz.) H.Ohba | 111 |
| *Castanea crenata* Siebold et Zucc. | 86 |
| *Quercus acutissima* Carruth. | 30 |
| *Zelkova serrata* (Thunb.) Makino | 7 |

19], namely

$$\log_e(Y_i) \quad \sim \text{Normal}(\mu_{s[i]}, \sigma^2_{s[i]}), \tag{1}$$

where $\mu_{s[i]}$ denotes the mean, and $\sigma_{s[i]}$ denotes the standard deviation for stand $s[i]$ containing the $i$-th individual. We excluded censored data from the model analysis, as the detection limits of such data were indefinite and varied depending on the measurement time.

We further assumed that $\mu_s$, the mean for stand $s$, was composed of the overall mean $\bar{\mu}$ and a random effect for the stand $\epsilon_{M[s]}$,

$$\mu_s \quad = \bar{\mu} + \epsilon_{M[s]}$$
$$\epsilon_M \quad \sim \text{Normal}(0, \sigma^2_M).$$

where $\sigma_M$ denotes the standard deviation of $\epsilon_M$.

We constructed two models predicated on distinct methodologies for formulating the standard deviation, denoted as $\sigma_s$, as outlined in Eq 1 (Table 2).

1. $\sigma_s$ is constant for all stands, namely $\sigma_s = \bar{\sigma}$.

2. $\sigma_s$ can vary due to a random stand effect, namely $\sigma_s = \bar{\sigma} \exp(\epsilon_{V[s]})$.

The random stand effect $\epsilon_V$ was assumed to follow a normal distribution, and can be represented as follows:

$$\epsilon_V \quad \sim \text{Normal}(0, \sigma^2_V).$$

The prior distribution of the overall mean was defined as a vague prior as follows:

$$\bar{\mu} \sim \text{Normal}(0, 100^2).$$

The priors of the standard deviations ($\bar{\sigma}$, $\sigma_M$, and $\sigma_S$) were defined as weakly informative priors [20, 21], as follows:

$$\sigma \sim \text{HalfNormal}(10^2).$$

The parameters in the models were estimated using a Markov chain Monte Carlo (MCMC) approach. The MCMC calculation was conducted using Stan 2.32.2 [22] via the CmdStanR 0.5.3 interface [23] for R [24]. We obtained the MCMC samples from four independent Markov chains, each consisting of 8,000 MCMC iterations after a warmup stage comprising 8,000 iterations. We then evaluated the convergence using the value of $\hat{R}$, or the Gelman-Rubin

**Table 2. Models to be compared.**

| Model | Structure |
|---|---|
| Model 1 | $\log_e(Y_i) \sim \text{Normal}(\mu_{s[i]}, \sigma^2_{s[i]}), \mu_s \sim \text{Normal}(\bar{\mu}, \sigma^2_M),$ |
| | $\sigma_s = \bar{\sigma}$ |
| Model 2 | $\log_e(Y_i) \sim \text{Normal}(\mu_{s[i]}, \sigma^2_{s[i]}), \mu_s \sim \text{Normal}(\bar{\mu}, \sigma^2_M),$ |
| | $\sigma_s = \bar{\sigma} \exp(\epsilon_{V[s]}), \epsilon_V \sim \text{Normal}(0, \sigma^2_V)$ |

$Y_i$: measured value of the radiocesium activity concentration in the growing shoots (Bq kg$^{-1}$) of the $i$-th individual, $\mu_{s[i]}$: mean of $\log_e(Y_i$ in the stand $s[i]$ containing the $i$-th individual, $\sigma_{s[i]}$: standard deviation of $\log_e(Y_i$ in the stand $s[i]$ (fixed in model 1), $\bar{\mu}$: overall mean of $\log_e(Y_i$ in the study area, $\bar{\sigma}$: overall standard deviation, $\epsilon_V$: random stand effect on $\sigma_s$, $\sigma_M$: standard deviation of the random effect $\epsilon_M$, $\sigma_V$: standard deviation of the random effect $\epsilon_V$.

statistics [25, 26] and a visual census of the degree of chain mixture. The Stan model code is included in the supporting information (S1 and S2 Files).

In addition, we also calculated the widely applicable information criterion (WAIC) [27] values for each model. The WAIC values were calculated under the stipulation of forecasting the radiocesium activity concentration in growing shoots in a new stand. Analogous to the Akaike information criterion (AIC), models characterized by smaller WAIC values are anticipated to exhibit superior mean predictive performance in the context of of model selection. Moreover, WAIC can be applied to complex hierarchical Bayesian models.

We fitted both models to the data containing only *Q. serrata* and selected the optimal model after consideration of WAIC values and model simplicity.

**Model for analyzing all data.** We then constructed a model to analyze the dataset comprising all five species that was based on the previously selected optimal model. In this model, we incorporated the origin of each individual (i.e., coppice sprout or plantation) as an explanatory variable, as well as an additional random species effect. The model selection affected whether the standard deviation, $\sigma_{s[i]}$ was uniform or varied across different stands.

$$\log_e Y_i \sim \text{Normal}(\mu_{s[i]}, \sigma^2_{s[i]}) \tag{2}$$

$$\mu_s = \bar{\mu} + \beta O + \epsilon_M + \epsilon_S \tag{3}$$

$$\epsilon_M \sim \text{Normal}(0, \sigma^2_M) \tag{4}$$

$$\epsilon_S \sim \text{Normal}(0, \sigma^2_S). \tag{5}$$

where the variable $O$ denotes the origin of the individual (0: sprout, 1: plantation), and $\beta$ is the coefficient of $O$. $\epsilon_M$ is a random stand effect, and $\sigma_M$ is the standard deviation thereof. Similarly, $\epsilon_S$ denotes a random species effect, and $\sigma_S$ is its standard deviation. Although *Z. serrata* had a small sample size, incorporating the random species effect facilitates estimates of individual species by "borrowing strength" [28].

For the parameter estimation, the model was fitted to the whole dataset (S1 Table), i.e. the data for all five species (40 stands and 418 individuals), using Stan 2.32.2. The code for the Stan model is given in the supporting information S3 File. Parameters used in the models are summarized in Table 3.

**Table 3. List of parameters.**

| Name | Description |
| --- | --- |
| $\bar{\mu}$ | Overall mean of log-transformed radiocesium activity concentration |
| $\mu_s$ | Mean of log-transformed radiocesium activity concentration at stand $s$ |
| $\beta$ | Coefficient of the trunk origin (0: sprout, 1: plantation) |
| $\epsilon_{M[s]}$ | Random stand effect on the mean of log-transformed radiocesium activity concentration at stand $s$ |
| $\epsilon_{V[s]}$ | Random stand effect on the standard deviation of log-transformed radiocesium activity concentration at stand $s$ |
| $\epsilon_S$ | Random species effect on the mean of log-transformed radiocesium activity concentration |
| $\bar{\sigma}$ | Overall standard deviation of log-transformed radiocesium activity concentration |
| $\sigma_s$ | Standard deviation of log-transformed radiocesium activity concentration at stand $s$ |
| $\sigma_M$ | Standard deviation of the random stand effect $\epsilon_M$ |
| $\sigma_V$ | Standard deviation of the random effect $\epsilon_V$ |
| $\sigma_S$ | Standard deviation of the random species effect $\epsilon_S$ |

**Prediction interval of the measurements of radiocesium activity concentration for a new stand.** Using the selected model with all data, we conducted a simple simulation to predict the radiocesium activity concentration in growing shoots in a new stand. In this simulation, we calculated the relationship between the sample size and the predicted value of the radiocesium activity concentration of growing shoots in the study area.

In addition, we estimated the relationship between the sample size and the standard deviation. Suppose a situation in which the radiocesium activity concentration is measured in $n$ individuals from a new stand.

Let $Y_{\text{new}}$ be the radiocesium activity concentration of the growing shoots in the new stand, and $y_{\text{new}} = \log_e(Y_{\text{new}})$. Namely,

$$y_{\text{new}} \sim \text{Normal}(\mu_s, \sigma_s^2). \tag{6}$$

The mean concentration $\mu_s$ follows a normal distribution with an overall mean of $\bar{\mu}$ and a variance of $\sigma_{\text{M}}^2$,

$$\mu_s \sim \text{Normal}(\bar{\mu}, \sigma_{\text{M}}^2). \tag{7}$$

As described above (Table 2), the standard deviation $\sigma_s$ depends on the model. If model 1 is selected, then $\sigma_s$ is a constant:

$$\sigma_s = \bar{\sigma}$$

Otherwise, if model 2 is selected, then $\sigma_s$ varies between stands:

$$\sigma_s = \bar{\sigma} \exp(\epsilon_{\text{V}}[s]).$$

From Eqs 6 and 7,

$$y_{\text{new}} \sim \text{Normal}(\bar{\mu}, \sigma_s^2 + \sigma_{\text{M}}^2).$$

Given the measurements $Y_1, Y_2, \ldots, Y_n$ (Bq kg$^{-1}$), we denote their logarithm values as $y_i = \log_e(Y_i)$, and $y_{n+1}$ should be distributed as follows [29]:

$$y_{n+1} \sim \text{Normal}(\mu_n, \sigma_s^2 + \sigma_n^2) \tag{8}$$

where the expected value of $\mu_n$ after $n$ individuals were measured is given by

$$\mu_n = \frac{\frac{1}{\sigma_{\text{M}}^2}\bar{\mu} + \frac{1}{\sigma_s^2}\sum_{i=1}^n y_i}{\frac{1}{\sigma_{\text{M}}^2} + \frac{n}{\sigma_s^2}},$$

and the expected value of $\bar{\mu}_n$ is

$$\bar{\mu}_n = \frac{\frac{1}{\sigma_{\text{M}}^2}\bar{\mu} + \frac{n}{\sigma_s^2}\mu_s}{\frac{1}{\sigma_{\text{M}}^2} + \frac{n}{\sigma_s^2}}, \tag{9}$$

In addition, $\sigma_n$ can be calculated as follows [29],

$$\frac{1}{\sigma_n^2} = \frac{1}{\sigma_{\text{M}}^2} + \frac{n}{\sigma_s^2}.$$

We calculated the mean $\mu_n$ for n = 1, ..., 10, under the assumption that the true value of the stand radiocesium activity concentration is 10 Bq kg$^{-1}$. We also calculated the standard deviation in Eq 8, $\sqrt{\sigma_s^2 + \sigma_n^2}$.

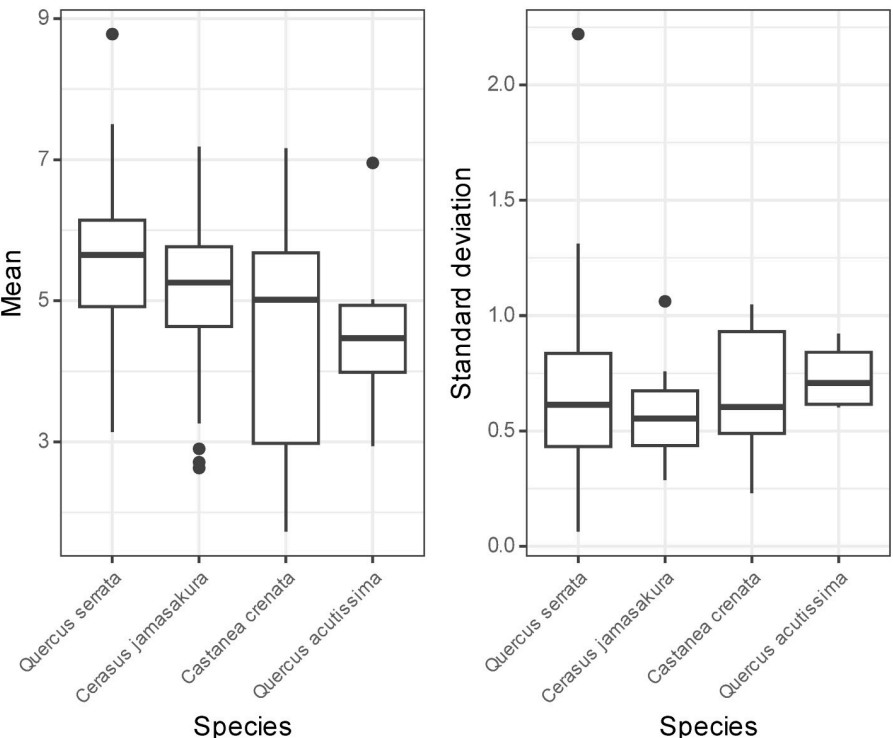

**Fig 1. Stand-level mean (left) and standard deviation (right) of log-transformed values of radiocesium activity concentration measurements in the growing shoots obtained for each species.** Values larger than the third quartile + 1.5 × interquartile range (IQR) or smaller than the first quartile—1.5 × IQR were treated as outliers. *Zelkova serrata* is not shown because the species occurred in only two stands.

Using the above relationships, we inferred how many individuals are required to adequately evaluate the expected value and error of the radiocesium activity concentration in a new stand.

## Results

### Radiocesium activity concentration measurements

The radiocesium activity concentration in the growing shoots ranged from 4.91 to $1.13 \times 10^4$ Bq kg$^{-1}$, and the mean and median were $4.65 \times 10^2$ and $2.02 \times 10^2$ Bq kg$^{-1}$, respectively. The stand-level mean of the log-transformed values of the radiocesium activity concentrations ranged from 1.73 to 8.78, and their mean and median was 5.08 and 5.29, respectively (Fig 1). The standard deviation in the stands ranged from 0.33 to 1.09 and the mean was 0.70. It should be noted that 7.7% of the total individuals surveyed were excluded from the analysis for the average value, as their samples could not be quantified due to low radioactivity levels; for the standard deviation, the value might therefore be larger if the excluded samples were included.

### Model selection

The results of the MCMC calculations for *Quercus serrata* showed that $\hat{R}$ values were smaller than 1.1 for all of the parameters, and that the Markov chains appeared to be well mixed. We therefore considered that the Markov chains converged successfully to stable distributions.

Table 4 shows the result of the model selection. Since there was only a small difference in the WAIC values between the two models (499 and 498), and because the values might vary

**Table 4. Comparison of models (posterior means of the parameters and WAIC).**

| Model | $\bar{\mu}$ | $\bar{\sigma}$ | $\sigma_M$ | $\sigma_V$ | WAIC |
|---|---|---|---|---|---|
| Model 1 | 5.62 | 0.74 | 1.09 | | 499 |
| Model 2 | 5.64 | 0.70 | 1.08 | 0.26 | 498 |

slightly due to the sampling error, we adopted the simpler model, Model 1. Consequently, the standard deviation could be perceived as uniform across the stands, thereby dismissing the variations intrinsic to individual stands in predicting the radiocesium activity concentrations.

## Parameter estimates for the whole dataset

Since the results of the model selection allowed us to treat the standard deviation $\sigma_s$ as being uniform among the stands, we let $\sigma_s = \bar{\sigma}$ and estimated the parameters of the model (Eqs 2–5) using MCMC.

Table 5 shows a summary of the posterior distributions for the parameters. The posterior mean of $\bar{\sigma}$, which is the standard deviation of radiocesium activity concentration in the growing shoots, was 0.74 on a logarithmic scale, and the standard deviation was 0.03. The value of 0.74 on the logarithmic scale translates to a geometric standard deviation of 2.09. Using this value, we can expect that approximately 95% of the measurement values are within an 18-fold ($\exp(2 \times 1.96 \times 0.74) = 18$) on the usual linear scale, excluding the effects of the species and origin. The posterior mean of the coefficient $\beta$, which represents the origin of the individuals, was estimated as -0.39, with the entire 95% confidence interval exhibiting negative values. The negative value indicates that the individuals from plantations would have a lower radiocesium activity concentration than the individuals produced from sprouts, and that the log-transformed mean concentration would decrease from 4.92 to 4.53. The posterior mean of $\sigma_M$, which is the standard deviation of the random stand effect on the log-transformed mean concentration, was 1.33, and larger than that of $\sigma_S$, 0.62, which is the standard deviation of the random species effect on the log-transformed mean concentration.

Fig 2 shows the posterior distributions of the random species effect. *Q. serrata* had a relatively high value while that of *Zelkova serrata* was lower, corresponding the log-transformed mean (Fig 1). Values in the figure are shown using a logarithmic scale; consequently, -1 on this scale is equivalent to a 0.37-fold decrease compared to the overall mean radiocesium activity concentration of the growing shoots on a linear scale. In the same manner, -0.5 on this scale is equivalent to a 0.61-fold decrease, while 0.25 and 0.5 are equivalent to increase of 1.6- and 2.7-fold, respectively.

Fig 3 shows the relationship between the number of measurement individuals ($n$) and the expected value of the log-transformed radiocesium activity concentration ($\log_e(\mu_n)$, assuming a true mean of 10 Bq kg$^{-1}$. This figure omits the random species effect and assumes the origin of the stems to be sprouts. It shows that the expected value progressively approximates the true

**Table 5. Summary of the posterior distribution of parameters for all species in the model.** SD: standard deviation.

| Parameter | Mean | SD | 2.5% quantile | 97.5% quantile |
|---|---|---|---|---|
| $\bar{\mu}$ | 4.92 | 0.41 | 4.06 | 5.65 |
| $\beta$ | -0.39 | 0.11 | -0.61 | -0.17 |
| $\bar{\sigma}$ | 0.74 | 0.03 | 0.69 | 0.79 |
| $\sigma_M$ | 1.32 | 0.17 | 1.04 | 1.71 |
| $\sigma_S$ | 0.62 | 0.44 | 0.19 | 1.76 |

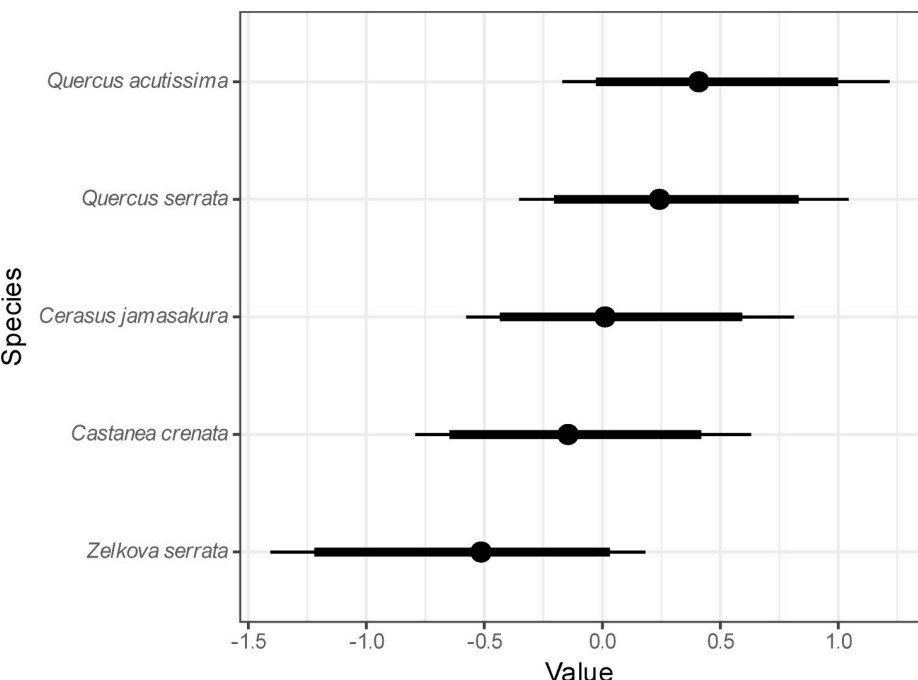

**Fig 2. Posterior distribution of the random species effect (on a logarithmic scale).** Dots denote medians, thick lines denote 90% credible intervals, and thin lines denote 95% credible intervals.

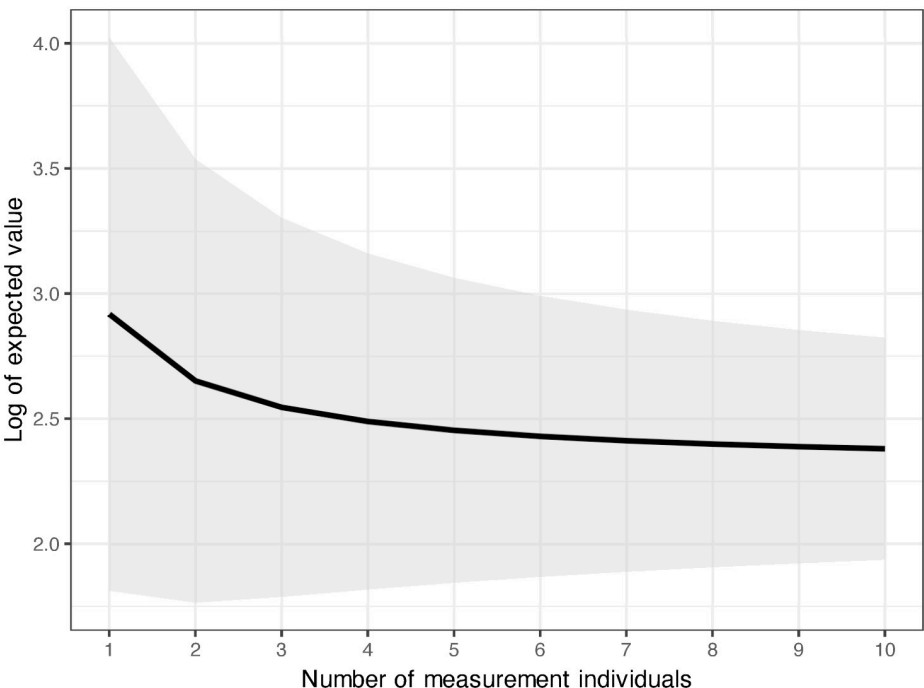

**Fig 3. Relationship between the number of measurement individuals and the expected value of the log-transformed radiocesium activity concentration when the true mean is 10 Bq kg$^{-1}$ (log$_e$10 = 2.30).** The curve denotes the mean, and the dark region denotes the 95% prediction interval of the expected value.

value as the number of measurements increases. Further, this relationship holds for other true values.

As a result, the standard deviation of the predicted radiocesium activity concentration on a logarithmic scale within a stand is given as

$$
\begin{aligned}
\sqrt{\sigma_s^2 + \sigma_n^2} \quad &= \sqrt{\sigma_s^2 + \frac{1}{\frac{1}{\sigma_M^2} + \frac{n}{\sigma_s^2}}} \\
&= \sqrt{\bar{\sigma}^2 + \frac{1}{\frac{1}{\sigma_M^2} + \frac{n}{\bar{\sigma}^2}}}.
\end{aligned}
\tag{10}
$$

## Discussion

In this study, we employed a Bayesian approach using MCMC methods to estimate parameters in the mixed-effect models. Using WAIC values facilitated the selection of a model for predicting radioactivity concentrations in the developing shoots of coppice hardwoods. In addition, the Bayesian approach facilitated straightforward management of uncertainties [29].

### Variation in radioactivity concentrations within a stand

Within the context of food safety, the public has a strong interest in radiocesium contamination of agricultural and forest products [7]. Local government bodies measure the radioactivity concentrations of foods and items utilized for food production, such as mushroom bed logs in the forestry sector, to prevent the distribution of foods and items that have radioactivity concentrations that exceed standard limits. Consequently, understanding the frequency distribution characteristics of the radioactivity concentration in forest products obtained under certain conditions is very important for the effective management and monitoring of these resources. In this study, we investigated the variability in radiocesium activity concentrations in growing shoots of hardwood species, which has been utilized to establish indices of radiocesium absorption [10, 11], with a particular focus on the hardwood bed logs that are commonly used in mushroom cultivation. The radiocesium activity concentration in the forest products has been demonstrated to follow a log-normal distribution [15, 19]. After the two largest nuclear power plant accidents to date, Chornobyl and Fukushima, radioactivity concentrations in many agricultural and forest products have been measured, and transfer factors (*TF*) and aggregated transfer factors (*T*ag) have been reported [30–32]. However, few studies have examined the variability—characterized by variance or standard deviation—in radioactivity concentrations of vegetation samples after log-transformation to attain normality. Khomutinin et al. [19, 33] analyzed large systematic grid sampling data, with variations in the dimensions of sampling units and increments (distances between the grid nodes). In their research on vegetation, they found that sampling independence in the radiocesium activity concentration could be achieved by sampling over distance of 8–10 m in stands with uniform soil contamination [15]. Their findings were reflected in the sampling guidelines recommended and published by the International Atomic Energy Agency (IAEA) [16]. Khomutinin et al. [19] also reported that the log-transformed standard deviation of radioactivity concentration by $^{137}$Cs sampled from 13 sites in Ukraine ranged from 0.22 to 0.62, and the mean and standard deviation was 0.38±0.11. Their sampling sites covered a variety of different land use types including cultivated lands.

Compared to the study in Ukraine, in this study, we estimated the standard deviation within 40 individual survey stands, focusing on the growing shoots (N = 5) of five hardwood species of forests in Fukushima. The range in the radiocesium activity concentrations showed a marked discrepancy between soil and growing shoots; while the range in the soils was relatively stable with a 10-fold difference, that in the growing shoots showed fluctuations that ranged up to 1000-fold [9]. Based on the WAIC values (Table 4), we could select the simpler model with a constant standard deviation, instead of the model with a varying standard deviation among stands. Thus, we could treat the variability of radiocesium contamination of forest products as uniform irrespective of stands, as shown in crops and grasses [19]. The standard deviation of log-transformed concentration within each stand ($\bar{\sigma}$) was estimated to be 0.74 as the posterior mean, and its standard deviation was estimated to be 0.03 (Table 5).

However, the within-stand standard deviation observed in this study and the study in Ukraine (0.38) [19] showed a two-fold difference. In Japan, woodlands are typically situated on steep undulating slopes, and the topographical characteristics of an area can change markedly within a relatively small area. We consider that these variations in the terrain may affect the soil properties. Furthermore, in terms of sampling methods, soil type, land usage, plant species, etc., numerous differences exist in the dataset used in the present study. Consequently, the factor that caused the two-fold difference between the present study and the preceding study in Ukraine was not specified. The standard deviations in the radioactivity concentration in vegetation samples are fundamental data that are essential for establishing radiation protection standards. Further studies are therefore required to clarify the principal factors that influence representative values and to comprehensively understand the prevalence of associated uncertainties.

In this study in Fukushima, we used sampling areas spanning several tens of square meters. To achieve statistical independence in sampling, Barnekow et al. [16] proposed that the center-to-center distance of sampling points for vegetation samples should not be less than 8–10 m. We consider that our sampling protocol satisfied this criterion in most cases. In contrast, in the survey of Khomutinin et al. [19], sampling grids ranged from 1 to 10 m in grid size, with 7 of the 13 surveyed sites having a grid size of 2.5 m or less. We considered the possibility that differences in sampling distance (increments) may have affected the differences observed in standard deviations, but the standard deviation in Khomutinin et al. [19] did not appear to be affected by the difference in grid size. All of the study sites in Khomutinin et al. [19] were located in crop fields and former agricultural fields and did not include forests; this is a notable difference between the datasets used in their study and this study. It is possible that forests show a greater variability of radiocesium activity concentrations in vegetation samples. For example, our Fukushima survey was conducted in mountainous areas with slopes of 6 to 30 degrees (11 to 73% gradient) [9], wheres all. 13 study sites in their Ukraine survey were located on flat terrain (confirmed on Google Maps from latitude and longitude). These differences may have accounted for the differences observed in the variability of radiocesium activity concentrations between these studies.

## Factors affecting the radioactivity concentration

The species random effect was larger in *Q. serrata*, which is most commonly used for bed logs; the posterior mean was 0.44. The random effect for *Q. acutissima*, another major hardwood species that is used for mushroom logs, was lower than that for *Q. serrata* (Fig 2); the posterior mean was -0.14. The difference in the posterior mean of the random effect was 0.57, which corresponds to a 1.8-fold difference when converted to a linear scale. The lowest value was observed in *Z. serrata*; the posterior mean was -0.55 or 0.37-fold of that of *Q. serrata* when

converted to a linear scale. The difference between the maximum and minimum mean values for the five tree species was 2.7-fold. Ipatyev et al. [34] reported a minimum 10-fold difference in $T$ag values ($m^2$ $kg^{-1}$) for needles and leaves of five tree species in Belarus, with a range from $1.6 \times 10^{-3}$ for birch to $3.8 \times 10^{-2}$ for pine. However, Calmon et al. [35] examined the $T$ag of trees after the Chornobyl nuclear accident and found that the influence of tree species exerted only a minor influence on these values, affecting them by a factor of 2 to 3. The present study also suggested that the interspecific difference in radiocesium absorption from the soil was small.

Furthermore, the data suggest a disparity in radiocesium contamination levels contingent on the origin of the trunk, with stems that originated from sprouts exhibiting a greater degree of contamination by radiocesium than stems that originated from plantations. In this study, the ages of the measured stems ranged from one year to a maximum of seven years. Therefore, the differences in radiocesium absorption among species and the origin of the stems may differ from those of stems aged around 20 years old, which are best suited for use as bed logs. Long-term monitoring is therefore considered necessary in order to asses whether the observed relationships remain consistent into the future.

The stand-level mean radiocesium concentrations observed in this study ranged from 1.7–8.8 in log-transformed values (Fig 1). However, since 7.7% of the total number of individuals were excluded due to their low radioactivity levels, it is plausible the true distribution lies more towards the left tail.

Kanasashi et al. [9] pointed out that the amount of exchangeable potassium in the surface soil largely influenced the radiocesium concentrations in the growing shoots. In comparison, although species and origin differences also affect radiocesium concentrations in the growing shoots, their impact is markedly diminished compaired to the growing environment. These findings are consistent with a review [35], where post-Chornobyl nuclear accident analysis of $T$ag values for trees showed a pronounced influence of soil type, quantified as a level of 100, whereas the effect of tree species was 2. Given this large variability in the mean values, it is important to investigate the radiocesium activity concentrations across stand to determine if the forest owners can resume the bed log production for mushrooms.

### Required sample size for appropriate estimation of radiocesium activity concentration

The objective of this study, evaluating the radiocesium activity concentrations of growing shoots in the coppices, is to protect communities from radiation exposure. This is particularly pertinent given the regulations that have been imposed on forest products used in food production, such as mushroom bed logs [7]. Assuming a normal distribution for the log-transformed radioactivity concentration estimates, the two parameters, i.e., the mean and standard deviation, are required to facilitate the accurate estimation of the proportion of the bed logs produced in a stand, which would adhere to the stipulated radioactivity regulation thresholds. Importantly, these parameter estimates have uncertainties. In this study, we showed that the within-stand standard deviation of the log-transformed radioactivity concentration of growing shoots of coppice woods could be regarded as uniform among stands and that the mean value was 0.74 (Table 5). The sample mean value of the log-transformed radioactivity concentration of growing shoots has been reported to vary markedly among stands [9]. Fig 3 shows the relationship between the number of measurement individuals ($n$) in a stand and the expected value of the log-transformed radiocesium activity concentration for the case where true mean is 10 Bq $kg^{-1}$ (2.30 in the log-transformed value). When the number of measurements is limited, the expected value approaches toward the overall mean of 4.92; however, with an increase in the data points, the curve progressively approximates

the true value, and the slope becomes gentler. Thus, the expected value would be 2.92 in the case where $n = 1$, 2.46 in the case where $n = 5$, and 2.38 in the case where $n = 10$ (Eq 9, Fig 3). Therefore, as the number of measurement individuals increases, the standard deviation of the predicted radiocesium activity concentration (log-transformed) within a stand diminishes; the standard deviation would be 0.98 in the case where $n = 1$, 0.80 in the case where $n = 5$, and 0.77 in the case where $n = 10$ (Eq 10). While the rate of decrease in the values slows down, the measurement cost increases in proportion to the number of individuals measured. Considering the tendency of decreasing standard deviation and the delicate balance between precision and cost, we propose that sampling five individuals is sufficient to estimate the radiocesium activity concentration in a coppice stand. This aligns with the recent guidelines issued by the International Atomic Energy Agency (IAEA) for soil and vegetation sampling [16]. However, it should be noted that the standard deviation of radioactivity concentrations in trees in Fukushima is approximately twice as large as observed in the previous study in Ukraine [19].

## Conclusion

In predicting radiocesium activity concentration in the growing shoots of hardwood species in forests exposed to radioactive fallout, the variance of the log-transformed measurement values within a stand can be assumed to be invariant irrespective of the stands. In this study, we found the mean standard deviation to be 0.74, a value that remained constant despite variations in the mean value of radiocesium activity concentration across stands and species. Furthermore, the findings showed that measurements from approximately five individuals would be sufficient for estimating the mean radiocesium activity concentration in a stand. The variance and mean of vegetation samples are fundamental parameters used in radiation protection measures, such as setting standard limits for radiation protection and determination surveys based on reference values. Numerous vegetation surveys have been conducted in forests, agricultural lands, and grasslands after the Fukushima and Chornobyl nuclear accidents. The statistical description and understanding of radioactive contamination in terrestrial plant ecosystems through integrated analysis of these data is an important topic that required further work.

## Supporting information

**S1 Table. Data file of the radioactivity concentration in the growing shoots of five hardwood species in Miyakoji.** The significant digits of the measurements are three.
(CSV)

**S1 File. Stan model code file for model 1.**
(TXT)

**S2 File. Stan model code file for model 2.**
(TXT)

**S3 File. Stan model code file for the model with all species.**
(TXT)

## Acknowledgments

The authors thank the Fukushima Chuo Forestry Co-operative for supporting this study. The authors thank Mr. Kentaro Matsuura (HOXO-M Inc.) for assistance with developing the Stan model incorporating the WAIC calculations. The authors also thank FORTE Science Communications (https://www.forte-science.co.jp/) for English language editing. This research was supported by the research program on development of innovative technology grants from the

Project of the Bio-oriented Technology Research Advancement Institution (BRAIN) (Grant Number: JPJ007097) and grants from the Forestry and Forest Products Research Institute (201901, 202203).

## Author Contributions

**Data curation:** Satoru Miura.

**Formal analysis:** Hiroki Itô.

**Investigation:** Hiroki Itô, Satoru Miura, Masabumi Komatsu, Tsutomu Kanasashi.

**Methodology:** Hiroki Itô, Satoru Miura, Masabumi Komatsu.

**Supervision:** Satoru Miura, Keizo Hirai.

**Writing – original draft:** Hiroki Itô, Satoru Miura.

**Writing – review & editing:** Hiroki Itô, Satoru Miura, Masabumi Komatsu, Tsutomu Kanasashi, Junko Nagakura, Keizo Hirai.

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
