## [Decision Letter · Decision Letter 0]

7 Nov 2023

Variability in radiocesium activity concentration in growing hardwood shoots in Fukushima, Japan

PONE-D-23-32470

Dear Dr. Itô,

We’re pleased to inform you that your manuscript has been judged scientifically suitable for publication and will be formally accepted for publication once it meets all outstanding technical requirements. 

Congratulations!!!

Reviewer #1 made a couple small suggestions that are not critical. I suspect you could accommodate this with small changes to the final text. These do not need review by me.

Kind regards,

Hanna Landenmark

Staff Editor

PLOS ONE

Additional Editor Comments (optional):

Reviewers' comments:

Reviewer's Responses to Questions

**Comments to the Author**

1. Is the manuscript technically sound, and do the data support the conclusions?

Reviewer #1: Partly

Reviewer #2: Yes

2. Has the statistical analysis been performed appropriately and rigorously? 

Reviewer #1: Yes

Reviewer #2: Yes

3. Have the authors made all data underlying the findings in their manuscript fully available?

Reviewer #1: Yes

Reviewer #2: Yes

4. Is the manuscript presented in an intelligible fashion and written in standard English?

Reviewer #1: Yes

Reviewer #2: Yes

5. Review Comments to the Author

Reviewer #1: Thank you for the opportunity to review your paper.

This paper identifies the variation in radiocesium radioactivity concentrations in logs contaminated by the FDNPP accident and estimated the sample size necessary to estimate these concentrations with sufficient accuracy. I have some concerns about some of expressions, which I hope will be corrected by the authors.

Major Issue

L351-384

The sentences are the main conclusion of the study. The authors actually investigated, however, it reads as if you took five samples are sufficient because of the previous study by Khomutinin et al. and the IAEA recommended. So, I recommend that to reinforce your argument by expressing why four samples is not enough, or how much uncertainty would be involved if five were sampled.

Minor Issue

L 7, 23

These sentences have no period.

L 83-97

How much the lower detection limit for the NaI(Tl) and Ge detectors, respectively?

L180-185, 368

Significant digits need to be accurate and consistent, i.e., 6 Bq have only one significant digit, 10265 Bq have five significant digits. What are significant digits for the actual measurement results? The same applies to the average. In my experience, the radioactivity standard gamma volume sources certified by the Japan Radioisotope Association have three significant digits at most.

Also, S1_data_analyzed.csv also does not have unified significant figures, but that is outside the scope of this review. So, I would like to suggest your voluntary review.

Reviewer #2: The authors report a numerical modeling study where they evaluated the variability in the radiocesium activity concentration of logs typically used for making beds for edible mushroom cultivation. The logs are from different study fields (stands) in the Fukushima Prefecture, Japan, in the vicinity of the 2011 Fukushima Nuclear Power Plant accident. Their results showed that the log-transformed radioactivity concentration of growing shoots of coppice woods could be regarded as uniform among stands with the mean standard deviation being 0.74. Their findings also showed that measurements from approximately five individuals would be sufficient for estimating the mean radiocesium activity concentration in a stand.

The study is well-planned and executed, the results clearly explained. The real strength of the report to me is the practicality of the study and the clarity of the delivery of the results. The authors put their study in context with the Chernobyl nuclear accident of 1986. While this study is not of mainstream interest and kind of focused on the Japanese food and forestry industry, it is a high-quality, careful work that is well presented.

I did not find any issues with the manuscript.

6. PLOS authors have the option to publish the peer review history of their article (what does this mean?). If published, this will include your full peer review and any attached files.

Reviewer #1: No

Reviewer #2: **Yes: **Tamas Varga

---

## [Editor Report · Acceptance letter]

13 Nov 2023

PONE-D-23-32470 

Variability in radiocesium activity concentration in growing hardwood shoots in Fukushima, Japan 

Dear Dr. Itô:

I'm pleased to inform you that your manuscript has been deemed suitable for publication in PLOS ONE. Congratulations! Your manuscript is now with our production department. 

Kind regards, 

on behalf of

Dr. Tim A. Mousseau 

Academic Editor

PLOS ONE